# The Interplay between the Cellular Response to DNA Double-Strand Breaks and Estrogen

**DOI:** 10.3390/cells11193097

**Published:** 2022-10-01

**Authors:** Lia Yedidia-Aryeh, Michal Goldberg

**Affiliations:** Department of Genetics, The Institute of Life Sciences, The Hebrew University of Jerusalem, Jerusalem 190401, Israel

**Keywords:** DNA damage response (DDR), estrogen, DNA double-strand breaks (DSBs), DSB repair, homologous recombination repair (HRR), estrogen receptor α (ERα)

## Abstract

Cancer development is often connected to impaired DNA repair and DNA damage signaling pathways. The presence of DNA damage in cells activates DNA damage response, which is a complex cellular signaling network that includes DNA repair, activation of the cell cycle checkpoints, cellular senescence, and apoptosis. DNA double-strand breaks (DSBs) are toxic lesions that are mainly repaired by the non-homologous end joining and homologous recombination repair (HRR) pathways. Estrogen-dependent cancers, like breast and ovarian cancers, are frequently associated with mutations in genes that play a role in HRR. The female sex hormone estrogen binds and activates the estrogen receptors (ERs), ERα, ERβ and G-protein-coupled ER 1 (GPER1). ERα drives proliferation, while ERβ inhibits cell growth. Estrogen regulates the transcription, stability and activity of numerus DDR factors and DDR factors in turn modulate ERα expression, stability and transcriptional activity. Additionally, estrogen stimulates DSB formation in cells as part of its metabolism and proliferative effect. In this review, we will present an overview on the crosstalk between estrogen and the cellular response to DSBs. We will discuss how estrogen regulates DSB signaling and repair, and how DDR factors modulate the expression, stability and activity of estrogen. We will also discuss how the regulation of HRR genes by estrogen promotes the development of estrogen-dependent cancers.

## 1. Introduction

Genomic instability, which is one of the hallmarks of cancer cells, plays a crucial role in cancer initiation and progression. A main cause for genomic instability is DNA damage [1,2,3,4,5]. DNA double-strand breaks (DSBs) are among the most deleterious type of DNA lesions [6]. DSBs occur constantly in cells by both endogenous sources and exogenous damaging factors [7,8]. The presence of DSBs in cells activates the DNA damage response (DDR), which is an extensive signaling network that includes DNA repair, cell cycle checkpoint activation, cellular senescence, and apoptosis. The DDR functions through the action of sensors, transducers, mediators and effectors. The main DSB sensor is the MRE11-RAD50-NBS1 (MRN) complex, which detects the breaks, binds them and activates the transducers of the cellular response to DSBs. The key DSB transducers are ATM and ATR, which are kinases that phosphorylate numerous DDR substrates and by that activate the DDR cascade. ATM is primarily activated by DSBs whereas ATR responds to DSBs, single-strand breaks (SSBs) and other lesions that interfere with replication. ATM exists in cells as an inactive dimer that upon DSB formation, in an MRN complex-dependent manner, undergoes auto-phosphorylation and dissociation, resulting in active monomeric ATM kinases that bind to DSBs. In the break sites, ATM phosphorylates, and thus activates, many DSB mediators and effectors, such as Chk1, Chk2, MDC1, BRCA1, MRE11, RNF8 and RNF168 [8,9,10,11,12,13,14].

DSB repair occurs by four DSB repair mechanisms, the major two repair pathways are the classical non-homologous end joining (NHEJ) and homologous recombination repair (HRR) pathways. The minor DSB repair pathways are alternative end joining (Alt-EJ) and single-strand annealing (SSA) error-prone repair pathways. Alt-EJ and SSA repair pathways initiate by the resection of the 5′ ends and are based on a short microhomology (Alt-EJ; 2–25 bp) or on a longer homology (SSA, >25 bp) [7,8,15,16]. NHEJ repair pathway involves direct ligation of broken ends and it is considered error-prone since it often requires processing of the broken ends prior to ligation. NHEJ initiates with the binding of the Ku70/Ku80 heterodimer to broken DNA ends. Next, Ku70/Ku80 heterodimer forms a complex with DNA-PKcs, which is the kinase catalytic subunit of the complex. The formed DNA-PK complex is activated at DSBs, resulting in the phosphorylation of members of the NHEJ machinery. Next, when required, the nuclease ARTEMIS is recruited to processes the DSBs, followed by DNA synthesis by DNA polymerase μ or λ. The final step is ligation, done by the XLF-XRCC4-DNA Ligase IV complex [17,18,19]. HRR DSB repair pathway initiates, similar to Alt-EJ and SSA, with a resection of the 5′ end. In HRR, homologous sequence, most often the sister chromatid, are used as a template for DSB repair and therefore HRR is restricted to the S and G2 phases of the cell cycle and is considered as an accurate repair pathway [17,20]. The resection process of HRR, which generates 3′ single-stranded DNA (ssDNA) overhangs, requires various proteins, including MRN complex (the DSB sensor), CtIP, BRCA1, BARD1, EXO1, DNA2, and BLM. The ssDNA is rapidly bound by the ssDNA-binding protein RPA following by replacement by the DNA recombinase RAD51, a process which requires BRCA2 and PALB2. RAD51 filaments, together with RAD54, take part in searching for and invading to the homologous DNA sequence. Next, RAD51 dissociates from the ssDNA, allowing a base-pairing between the invading and complementary donor strands. This is followed by a strand extension. The extended strand dissociates and anneals with the processed end of the non-invading strand on the opposite side of the break, and then invades to produce a double-Holliday junction. The last step of HRR is the junction resolve, which requires BLM and additional helicases, yielding both crossover or non-crossover recombinants [17,20,21].

DNA repair pathways are frequently compromised in cancer cells. Predominantly, HRR genes are associated with breast and ovarian cancers, which are common cancers among women in the western world. Hereditary breast and ovarian cancers result mainly from inherited mutations in HRR genes, when the most common inherited mutations are in BRCA1 or BRCA2 genes. However, many other HRR genes, such as PALB2, BLM, members of the MRN complex, and ATM have also been associated with this cancer [22].

The aetiology of estrogen-dependent cancers, which include breast, ovarian and endometrial cancers, is multi-factorial, and includes, apart from the genetic risk factors, estrogen. Estrogen is referred to as a group of sex steroid hormones that regulate the growth, development, and functions of the reproductive system in females. Apart from its reproductive functions, estrogen also plays a role in additional physiological processes, including processes involved in cardiovascular, neuronal and skeletal systems. 17β-estradiol (E2) is the most abundant and potent of the three natural estrogens. Notably, risk factors for breast cancer are strongly connected to estrogen-related pathways [23,24,25,26,27,28]. An increased exposure to estrogen due to early menarche, late menopause, and absence of childbearing augment breast cancer risk. Furthermore, aberrant hormone exposure, whether induced naturally or due to clinical administration, increase the risk of breast cancer [29,30,31,32]. Additionally, it was shown in female ACI (August/Copenhagen/Irish) rat model that treatment with physiological serum levels of estrogen leads to mammary gland tumors [33]. In this rat model, estrogen induced centrosome amplification, chromosomal instability and aneuploidy that triggered the development of breast cancer [34].

Estrogen executes its biological functions mainly by binding and activating several receptors, the most prominent are estrogen receptor (ER) α,ERβ and the recently identified G-protein-coupled ER 1 (GPER1). ERα and ERβ (hereinafter referred to as ERs) are intracellular receptors while GPER1, is a membrane bound receptor. GPER1 has a weaker binding affinity to estrogen compared to the ERs. The ERs are transcription factors that bind their target genes at estrogen response elements (EREs). ERα also associates with the plasma membrane where it activates non-nuclear signaling. ERα activates pro-proliferative signals, targeting genes that promote cell proliferation or inhibit apoptosis. ERβ has anti-proliferative effects and thus antagonizes ERα [35,36,37,38]. The interaction between estrogen and ERα triggers ubiquitination of the receptor. Monoubiquitination of ERα modulates its stability and transcriptional activity, while Lys48-linked polyubiquitination of the receptor targets it to proteasomal degradation [39]. Estrogen regulates the transcription of several DDR factors, mainly of HRR genes, it also regulates the stability and activity of numerous DDR factors. Moreover, several DDR factors modulate the expression of ESR1, which encodes ERα, as well as stabilize ERα and regulate ERα-mediated signaling [40,41,42,43,44,45,46,47,48,49,50,51,52,53,54,55,56,57,58,59].

In this review, we will discuss the link between estrogen signaling, DSB formation and DSB repair.

## 2. Estrogen Induces DNA Damage in Cells

Estrogen and estrogen-related pathways are associated with genomic instability. One of the mechanisms by which estrogen triggers genomic instability is the induction of DNA damage in cells during estrogen stimulation. Estrogen induces DNA damage in cells by several manners, all of which may result in DSBs, which may occur in estrogen-responsive genes of at global genomic regions (Figure 1):

### 2.1. DNA Lesions as Byproducts of Estrogen Metabolism

Estrogen metabolism results in the accumulation of reactive oxygen species (ROS) and nitric oxide moieties, leading to oxidative DNA damage in the cells [60]. ROS in cells may result in base modifications, SSBs or DSBs [61].

### 2.2. Induction of Mitochondrial ROS by Estrogen

Estrogen addition rapidly induces hydrogen peroxide based mitochondrial ROS. This activity of estrogen acts as a signal-transducing messenger that includes enhanced cell motility. The mitochondrial ROS formation triggered by estrogen occurs in epithelial cells. It depends on cell adhesion, the cytoskeleton, and integrins by it is independent of the ERs [62].

### 2.3. DSB Formation at the Promoters of Estrogen-Responsive Genes

Estrogen supplementation rapidly induces DSBs at the promoters of estrogen-responsive genes. These DSBs are required for the transcriptional activity of estrogen [63,64,65] and are repaired by HRR [63]. DSB formation by estrogen at the promoters of estrogen-responsive genes is dependent on the DNA cytosine deaminase APOBEC3B [64] and topoisomerase IIβ [63,65].

### 2.4. DSB Formation as a Result of R-Loops Formation by Estrogen

Addition of estrogen to cells leads to an upsurge in R-loop (DNA:RNA hybrid) formed at estrogen-responsive genes [66]. R-loop formation may result in DSBs via different mechanisms; the displaced single-stranded DNA can be converted to a DSB during DNA replication, the R-loop may be nucleolytic processed or cause replication fork stall which triggers DSB formation as well as by collision of the replication fork with a backtracked RNA polymerase [67]. R-loops at estrogen-responsive genes are enriched in estrogen-responsive genes. They are formed prior to the onset of DSBs, in cells that have not yet entered the S phase of the cell cycle. R-loop-dependent DSBs are induced in a replication- and transcription dependent manner [66].

In correlation to the findings that many of the estrogen-induced DSBs occurs at estrogen-responsive genes, it was shown that regions of estrogen-activated transcription are more frequently mutated in breast cancers [66].

## 3. Estrogen as a Regulator of the Cellular Response to DSBs

Estrogen, in parallel to activation of cell proliferation and DSB formation, modulates the expression and activity of numerus factors involved in the cellular response to DSBs.

### 3.1. Estrogen Modulates the Expression of DDR Factors

Supplementation of estrogen to ER-positive cells either suppresses or enhances transcription of numerous genes encoding key DDR factors.

#### 3.1.1. Estrogen Negatively Regulates the Expression of the Key DSB Transducer, ATM

ATM activates the DDR by phosphorylation of many DDR members [8,9,10,11,12,13]. Estrogen negatively regulates the expression levels of ATM in an ERα-dependent manner. The regulation of ATM by estrogen is indirect; Estrogen addition upregulates the expression of two miRNAs, miR-18a and -106a. These microRNAs downregulate the expression of ATM. In agreement with these findings, the expression of ATM is elevated in ERα-negative breast cancer tissues, where as the expression of miR-18a and -106a is significantly reduced in these tissues [35,68,69].

#### 3.1.2. Estrogen Positively Regulates the Expression of Key DSB Repair Factors

The expression of a large fraction of DSB repair genes is regulated by estrogen. Our analysis revealed that genes belonging to the KEGG HR pathway are enriched among genes that are upregulated by estrogen. When examining the percentage of the estrogen-upregulated genes among DSB repair genes we found that 31% and 24.7% of established HRR and NEHJ genes, respectively, are regulated by estrogen. Notably, the proportions of the genes that are regulated by estrogen among other repair pathways are lower (20.9% for base excision repair, 20.5% for mismatch repair, and 10.4% for nucleotide excision repair (NER)) [40].

Specifically, estrogen directly upregulates the expression of the key NHEJ player, the kinase DNA-PKcs. This positive regulation is direct and ERα-dependent [70]. Furthermore, estrogen is a master regulator of HRR gene expression (Table 1). It positively regulates mRNA levels of many HRR genes, including BRCA1 and BRCA2, which are frequently mutated in familial breast and ovarian cancers [41,52,53], BLM helices [54], CtIP [55,56] and all members of the MRN complex (MRE11, Nbs1 and RAD50) [40,57] (see Table 1 for additional DSB repair genes that are regulated by estrogen). In agreement, the increase in mRNA levels of DSB repair genes is in accordance with the increase in protein levels of the tested DSB repair factors (Table 1). Notably, regulation of the expression of HRR factors by estrogen is not depend on DSB formation. In addition, we found that levels of all tested mRNAs encoding HRR factors increase in estrogen-deprived cells following DSB induction (Table 1). The findings that estrogen increases the expression of HRR genes are in correlation with the results indicating a reduced expression of all members of the MRN complex in ERα-negative cells compared to ERα-positive cells [71]. Furthermore, the level of the phosphorylated form of BRCA1 (p-S988), which is the active BRCA1 form that triggers DSB repair via HRR [72], is reduced in ERα-negative cells compared to ERα-positive cells [71].

### 3.2. Non-Translational Regulation of DSB Repair by Estrogen

As mentioned above, estrogen modulates the mRNA levels of many DDR factors, most of which are HRR genes, via direct transcriptional regulation of these factors. However, estrogen can also regulate the stability of the HRR factor, BRCA2. Estrogen rapidly stabilization of BRCA2 protein levels in ER-positive breast cancer cells. is probably mediated by a CDK-dependent mechanism that involves phosphorylation of BRCA2 protein [74].

### 3.3. Estrogen Negatively Regulates the Activity of the DSB Transducer ATR and the Effector Kinase Chk1

Estrogen suppresses ATR activity without affecting its expression. The negative regulation of ATR activity by estrogen requires plasma membrane-localized ERα. Following DNA damage induction, there is an augmentation in the interaction between ATR and TopB1, which is correlated with ATR activity. Estrogen reduces the ATR-TopB1 interaction by triggering phosphorylation of TopBP1 by AKT. Phosphorylated TopBP1 will bind E2F1, resulting in reduced interaction between ATR and TopBP1 after DNA damage, leading to suppressed activation of the DDR by ATR.

Estrogen also inhibits the activity of the effector kinase Chk1 in a similar manner to that of ATR. Estrogen inhibits the interaction between Chk1 and Claspin, which is a coactivator of Chk1. Estrogen induces phosphorylation of Chk1 by AKT, and this reduces the Chk1-Claspin interaction, resulting in suppression of the DDR [69]. Notably, the expression of Chk1 is also down-regulated by estrogen addition [79].

Inhibition of the activity of ATR and Chk1 by estrogen results in impaired G2/M checkpoint activation, faulty DSB repair (see below) and reduction in p53 transcription, which may have an impact on the activation of the G1/S checkpoint [69].

Taken together, current data suggests that estrogen reduces the amounts and/or activity of factors that act upstream of the DDR; ATM, ATR and Chk1 [35,68,69,79], while increasing the amounts of factors that play a role in DSB repair pathways [40,41,52,53,54,55,56,57,70,74] (Figure 2). This implies that the negative regulation of ATM, ATR and Chk1 is required for the proliferation effect of the estrogen pathway, since reduced amounts of these key DDR kinases may facilitate to overcome cell cycle regulation. On the other hand, in order to ensure genomic stability of the proliferative cells, and to protect the cells from the DNA lesions induced by estrogen [63,64,65,66], the estrogen signaling pathway ensures rise in the levels of DSB repair factors, which is most likely correlated with more efficient DSB repair (Figure 2).

## 4. Regulation of the Expression and Activity of ERα by DSB Factors

Not only that estrogen regulates the DDR via modulating the amounts, stability and activity of members of the cellular response to DSBs, several members of the response control the expression, stability and activity of ERα.

### 4.1. BRCA1 Inhibits the Activity of ERα

BRCA1 suppresses ERα activity by different mechanisms (see below [42,43,44,58,59,71,80]). This inhibition is important for maintenance of genomic stability. For example, numerous mutations in the BRCA1 gene results in impaired ability to inhibit the transcriptional activity of ERα [59]. Additionally, there is an estrogen independent ERα-mediated transactivation only in BRCA1-deficient cells [80].

#### 4.1.1. BRCA1 Interacts with ERα

BRCA1 protein binds ERα. This interaction is estrogen-independent and it requires the amino-terminal region of BRCA1 and the conserved carboxyl-terminal activation function domain of ERα [42,59]. The interaction between BRCA1 and ERα occurs, at least in part, at EREs, where it is detected mainly prior to estrogen stimulation. Upon estrogen addition, there is a remarked reduction in the level of BRCA1 at EREs [80]. Notably, while BRCA1-ERα interaction is estrogen independent, it is increased following DSB induction [71]. The biological mechanism by which the BRCA1-ERα interaction inhibits the transcriptional activity of ERα is multifaceted, and this inhibition can be reversed by overexpression of p300, CBP [42], or cyclin D1 [81]. p300 and CBP, which are highly homologous acetyltransferases, interferes with BRCA1 inhibitory effect probably via acetylation of ERα [42,82,83], whereas cyclin D1 competes with BRCA1 for the binding of ERα [81].

#### 4.1.2. BRCA1 Downregulates p300 Expression and Reduces ERα Acetylation and Activity

p300 is a lysine acetyltransferase that acetylates histone and non-histone proteins to modulate transcription. It is a co-activator for ERα-signaling pathway; it acetylates the receptor and enhances its stability [82,83]. BRCA1 reduces p300 mRNA and protein levels, resulting in suppressed activity of ERα [42]. Indeed, BRCA1 regulates the acetylation of ERα in ER-positive cells and mutation of an acetylation motif in ERα confer resistance to repression of the receptor by BRCA1 [44].

#### 4.1.3. BRCA1 Monoubiquitinates ERα

ERα is monoubiquitinated by BRCA1. This ubiquitination event requires also BARD1, which is a protein that interacts with BRCA1 [43]. BRCA1 overexpression augments ERα monoubiquitination and reduces its acetylation. Thus, BRCA1 represses the activity of ERα, in part, by regulating the relative amount of its acetylation vs. ubiquitination [44].

BRCA1 suppresses the transcriptional activity meditated by ERα and many breast cancer-associated mutations of BRCA1 abolish or reduce its ability to repress the activity of the receptor. Since estrogen regulates the expression of BRCA1 gene, there is a negative feedback loop that modulates cell proliferation; Estrogen, via the activation of ERα, promotes proliferation and increases BRCA1 levels and BRCA1, in turn, inhibits ERα activity, resulting in reduced BRCA1 expression and cell proliferation. This feedback loop balances between the proliferative effect of estrogen and the requirement for maintenance of genomic stability. This is in line with BRCA1 being an estrogen-dependent cancer susceptibility gene, as loss of functional BRCA1 in cells tips the balance to proliferation and triggers tumorigenesis.

### 4.2. MDC1 Co-Activates ERα-Mediated Transcription

MDC1, is an upstream mediator of the DDR, which plays an early and crucial role in DDR activation [84]. MDC1 physically interacts with ERα. This interaction in augmented upon estrogen stimulation. MDC1 is recruited, together with ERα, to EREs of ERα-responsive genes and enhances ERα-mediated transactivation [45].

### 4.3. RNF8 and RNF168 Co-Activate ERα-Mediated Transcription

RNF8 and RNF168 are E3 ubiquitin ligases that transduce DSB signal and are essential for efficient DDR. RNF8 ubiquitinates histone H2A and H2AX around DSB sites and this ubiquitination event is amplified by RNF168 [85]. RNF8 interacts with ERα and monoubiquitinates it. This monoubiquitination event results in increasement in the stability of ERα as well as in enhancement of ERα-mediated transactivation [46]. RNF168 is recruited to the promoter of ERα and regulates ERα mRNA levels as well as ERα-mediated transcription [47].

### 4.4. DNA-PK Co-Activates ERα-Mediated Transcription

The complex DNA-PK is the key component of NHEJ repair. It consists of the heterodimer Ku70/Ku80 and the catalytic subunit DNA-PKcs. There is a positive feedback loop between DNA-PK and estrogen. DNA-PK enhances ERα-mediated transcription [86] and estrogen augments the expression of DNA-PKcs gene [70]. Ku70, Ky80 and DNA-PKcs interact with ERα, and this interaction is augmented in the presence of estrogen. DNA-PKcs phosphorylates ERα, and this phosphorylation event prevents polyubiquitination of the receptor, resulting in ERα stabilization. In parallel, DNA-PKcs supports ERα-mediated transactivation [86].

When comparing the interplay between ERα and BRCA1 to that of other DSB response factors, it appears that the relationship between ERα and BRCA1 is unique. While BRCA1 reduces the activity of the receptor, other analyzed DSB response factors, which act upstream to the DDR (RNF8, RNF168 and MDC1) or play a role in NHEJ (DNA-PK) enhance its activity. It will be interesting to reveal whether inhibition of ERα activity is unique to BRCA1 or common among HHR factors.

## 5. Estrogen and DSB Repair Efficiency

It is not clear how the effect of estrogen, which results in the upregulation of mRNA and protein levels of DSB repair pathway genes (Table 1), is translated into DSB efficiency. Notably, there is a controversy in the field regarding the effect of estrogen on DSB repair. We and others have shown that estrogen is required for intact DSB repair, since we found that DSB repair is impaired in ER-positive cells depleted for estrogen [40,70,71]. On the other hand, it was demonstrated that estrogen delays DSB repair [69] and specifically that the frequency of HRR is higher in ER-positive cells deprived of estrogen compared to cells re-supplemented with the hormone [77].

The dispute regarding the effect of estrogen on DSB repair may result from the different DNA-damaging agents used in the above studies. Studies that support a positive role for estrogen on DSB repair induced DSBs by ionizing radiation [70] or by DSB-inducing chemicals (neocarzinostatin [40] or cisplatin [71]). In contrast, studies that support a negative role for estrogen in DSB repair used ultra violet radiation, in which DSB are byproducts of lack repair of thymine dimers by NER [69] or by the restriction enzyme I-SceI [77]. In the latter study, the DR-GFP reporter cassette was utilized to determined HRR efficiency. In this system, I-SceI is present in cells for a prolonged time, and the DSB can be accurately repaired by NHEJ, resulting in several cycles of DSB formation and repair [87,88]. The finding that estrogen augments HRR levels [46,55], can explain the higher HRR efficiency obtained using the DR-GFP system in cells deprived of estrogen [77]; Induction of DSBs, due to expression of I-SceI, in cells deprived of estrogen, results in an immediate rise in mRNAs encoding HRR factors. These transcripts are translated to HRR factors throughout the process of cycles of DSB formation by I-SceI and repair by precise NHEJ, eventually resulting in more efficient HRR repair.

## 6. The Connection between Estrogen Regulation of HRR Genes and Estrogen-Related Cancer Development

Germline variants in numerus genes that play a role in the HRR pathway, such as BRCA1, BRCA2, ATM, BRIP1, NBS1, PALB2, RAD51C and RAD51D, lead to inherited susceptibility to breast, ovaries, endometrial, prostate, and pancreas cancers [22]. Of note, estrogen plays a role in the etiology of these cancers, as it drives tumor development and progression [23,24,25,26,27,28,89,90]. This suggests a tight crosstalk between estrogen and the HRR pathway. DNA damage is deleterious to all cells, but mainly to proliferating cells. Since proliferation increases DNA damage, the proliferating effect provided by estrogen via ERα signaling should be tightly regulated. On top of that, estrogen by itself induces DNA damage in cells [60,63,64,65,66], adding an additional level to the importance of a tight link between DNA repair and estrogen signaling. Fine-tuning of the ERα signaling cascade is achieved by a crosstalk between estrogen, ERα, and the cellular response to DSBs ensures maintenance of genomic integrity. The expression of many HRR genes is regulated by estrogen and DDR factors modulate the expression, stability and/or activity of ERα [40,41,42,43,44,45,46,47,48,49,50,51,52,53,54,55,56,57,58,59]. Thus, the crosstalk between estrogen and the HRR pathway ensures maintenance of genomic integrity.

We suggest a model that explains why mutations in numerus HRR pathway genes predispose to estrogen-related cancers (Figure 3); In healthy cells, there is a thigh regulation on the proliferation effect that estrogen imposes on cells by factors involved in the DDR. Estrogen augments the expression of HRR genes in proliferating cells, to overcome accumulation of DNA damage that is induced during estrogen stimulation. This is essential for maintaining genomic stability since DNA damage is detrimental to proliferating cells. Augmentation of HRR factors will eventually result in more efficient HRR. Efficient HRR is required in order to repair replication-induced DSBs that accumulate in the proliferative state as well as DSBs that are formed by estrogen. These DSBs can occur at random genomic sites following estrogen metabolism and mitochondrial ROS, at the promoters of estrogen-responsive genes, due to ER-mediated transcriptional activity or at estrogen-responsive genes as a result of R-loop formation [60,62,63,64,65,66].

In the case that a cell has a germline mutation in a gene encoding a member of the HRR pathway, the fine-tuning of the estrogen response will be impaired and the proliferative effect of ERα signaling will not be properly regulated. Estrogen will induce cell proliferation in these cells, and will augment the expression of HRR genes. However, since there is a germline mutation in one of the HRR genes, there will probably be insufficient activity of this factor, resulting in a faulty activation of the HRR pathway. On top of that, since estrogen supplementation induces DSBs at estrogen-responsive genes [63,64,65,66], and thus at genes encoding to HRR factors, the chances for additional mutations at HRR genes increases. Moreover, we found that in ER-positive cells, DNA damage augments levels of HRR factors in the absence of estrogen. Thus, similarly to what occurs due to the presence of estrogen, in cases of a germline mutation in a HRR gene, there will be no increased activity of the mutated HRR factor upon DSB induction. This will result in impaired DSB repair. The above will result in accumulation of mutations and chromosomal aberrations, which will trigger cancer formation (Figure 3). Indeed, BRCA1 mutations, as well as mutations in other HRR genes, predispose genomic instability and trigger cancer formation. Recently, it was shown, in a mouse model, that genomic instability in BRCA1 heterozygous mice appears already at 10.5-day embryo and progressively towards adulthood. Interestingly, accumulation of genomic instability is not linear and there are many dynamic changes during the embryonic stage [91].

## 7. Conclusions

Studies indicate that there is a tight crosstalk between estrogen, HRR factors and DSB formation and repair. Estrogen regulates the expression of HRR factors and the activity of upstream DDR factors. In addition, it has an impact on DSB repair. In turn, DDR factors modulate the signal of estrogen, by affecting the activity, expression and stability of ERα. The carcinogenic effect of estrogen, which triggers the development of several cancers, including, breast, ovaries, endometrial, prostate, and pancreas cancers place estrogen, puts the estrogen pathway as a central therapeutic target. Further uncovering the mechanisms that modulate the interplay between estrogen, DDR factors and DSB formation and repair are crucial for the development of therapeutic or preventative strategies targeting estrogen-related cancers. Furthermore, the interplay between ERβ and DDR factors must be further studied in order to get the full overview of the crosstalk between estrogen and the DDR.

## Figures and Tables

**Figure 1 cells-11-03097-f001:**
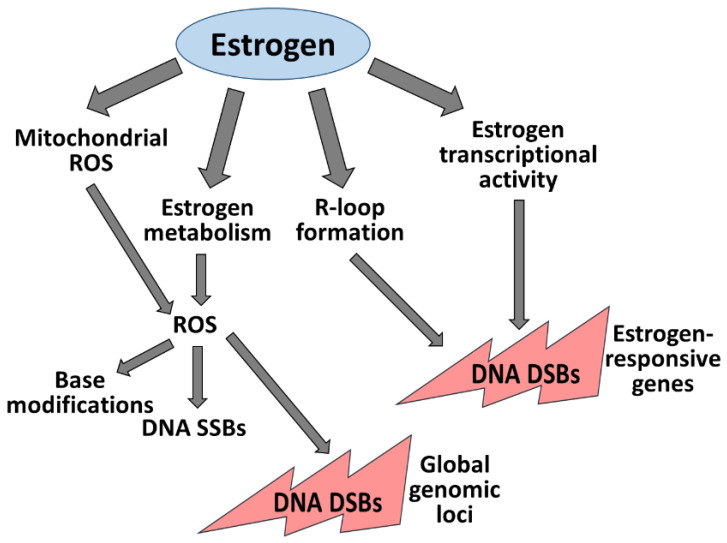
Estrogen induces DNA damage in cells. Estrogen prompts hydrogen peroxide based mitochondrial ROS and a bypass of estrogen metabolism is ROS. ROS leads to several types of DNA lesions, including base modifications, SSBs and DSBs. These DSBs are dispersed throughout the genome. Estrogen stimulus increases R-loop formation, which are processed into DSBs, in estrogen-responsive elements. Besides, estrogen addition induces DSBs at the promoters of estrogen-responsive genes.

**Figure 2 cells-11-03097-f002:**
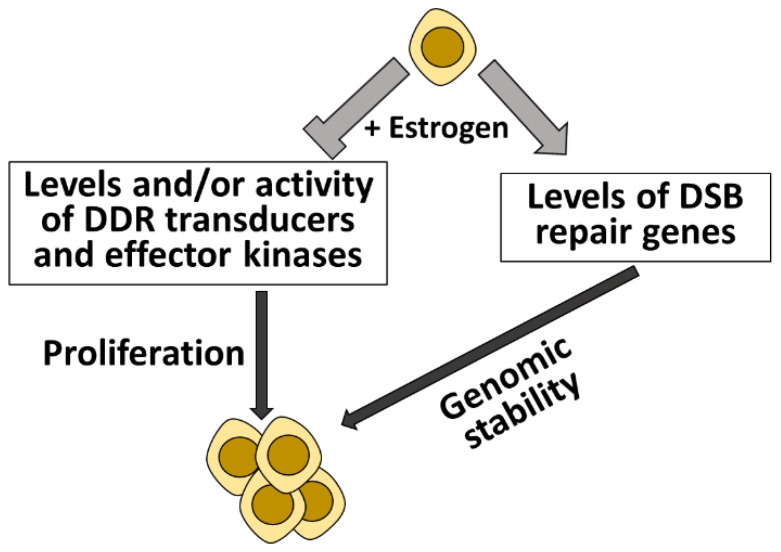
Estrogen regulates the level and activity of numerous DDR genes to allow cell proliferation and to ensure genomic stability. Addition of estrogen to cells results in reduced amounts ATM and Chk1. It will also reduce the activity of ATR and Chk1. These three kinases act upstream to the DDR, and their activity restrains cell proliferation. Thus, reduction in their level/amount will trigger cell proliferation. On the other hand, estrogen addition results in augmentation of genes directly involves in DSB repair, which will improve DSB repair and safeguard genomic stability of the proliferating cells.

**Figure 3 cells-11-03097-f003:**
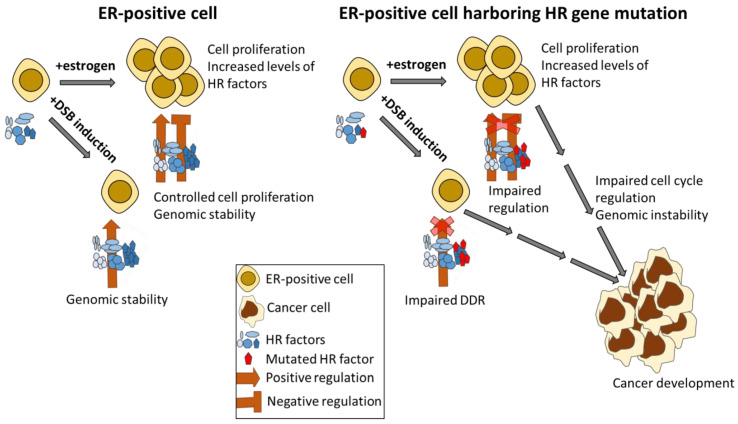
A model that depicts the interplay between estrogen, HRR factors, DNA damage and DDR and suggests how this interplay triggers estrogen-dependent cancer formation in cells harboring mutation in HRR genes. ER-positive cells contain a basal level of mRNA encoding HRR factors. Upon supplementation of estrogen or due to DSB formation, the level of these mRNAs is elevated. Addition of estrogen results in cell proliferation, which is tightly regulated, both positively and negatively, by DDR factors, to ensure controlled cell proliferation and genomic stability. Incensement in HRR factors due to DSB induction in ER-positive cells deprived of estrogen will upsurge the activity of the HRR pathway to ensure genomic stability. In cells harboring a mutation in one of the HRR genes, there is only a partial augmentation of HRR genes, resulting in insufficient activity of the HRR pathway. These will lead to impaired cell cycle regulation and genomic instability that may trigger cancer development.

**Table 1 cells-11-03097-t001:** A list of DSB repair genes that their mRNA levels are upregulated by estrogen.

DSB Repair Pathway	Estrogen Increases mRNA Level	Direct Binding of ERα to the Promotor	Estrogen Increases Protein Level	DSB-Induced in Estrogen-Deprived Cells
NHEJ	DNA-PKcs [70]	Yes [70]	Yes [70]	Not known
HRR	BRCA1 [41,52,53]	Controversy [41,52,53]	Yes [73]	Not known
	BRCA2 [41]	Not known	Yes [74]	Not known
	BLM [40,54]	Not known	Yes [54]	Yes [40]
	CtIP [40,55,56]	Not known	Not known	Yes [40]
	BARD1 [75]	Yes [75]	Yes [75]	Not known
	CtIP [40,55,56]	Not known	Not known	Yes [40]
	DNA2 [76]	Not known	Not known	Not known
	RAD51 [77]	Not known	Yes [77]	Not known
	RAD51C [78]	Not known	Yes [78]	Not known
	RAD54D/L [76]	Not known	Not known	Not known
	PALB2 [40]	Not known	Not known	Yes [40]
	MRE11 [40]	Not known	Not known	Yes [40]
	NBS1 [57]	Not known	In combination with DNA damage [57]	Not known
	RAD50 [40]	Not known	Not known	Yes [40]

## Data Availability

Not applicable.

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
