# Peer review of "The Interplay between the Cellular Response to DNA Double-Strand Breaks and Estrogen"

_cells, 2022, doi:10.3390/cells11193097_

Round 1
Reviewer 1 Report
The review by Yedidia-Aryeb and Golberg is an interesting work. It is clear, and the models are critically evaluated and discussed. It appears to be quite comprehensive. The authors could address two minor points:
- Mention the G protein-coupled estrogen receptor 1 as an additional member of the estrogen receptor family;
- Distinguish the few works that address the relationship between DNA damage response and estrogen in animal models.
Author Response
We would like to thank Reviewer#1 for his/her approval of our manuscript and for the excellent minor points.
- Mention the G protein-coupled estrogen receptor 1 as an additional member of the estrogen receptor family
In the revised manuscript we added the G protein-coupled estrogen receptor 1 as an additional member of the estrogen receptor family – it is mentioned both in the abstract and introduction part. There are no published studies that connect this receptor to DNA double-strand breaks, and therefore there is no additional relevant data for this review.
- Distinguish the few works that address the relationship between DNA damage response and estrogen in animal models.
We could not find many works that address the relationship between the DDR and estrogen in animal models, therefore we felt that a separate section on animal models will not provide enough information. However, we did add studies, made in animal model, to the different sections of the review and mentioned that they were performed in animal models.
Reviewer 2 Report
Aryesh and Goldberg have written a comprehensive and relevant review article discussing the diverse roles estrogen in DNA damage response and interplay between DDR and estrogen. The authors discusses how estrogen induce DNA damage, regulate expression of DNA repair factor, affect response to DNA damage and vice versa i.e. how DNA damage affect estrogen. The unique feature about the article is that no such articles has been published before that comprehensively cover different aspects of interplay between estrogen and DDR. The review covers major findings in the field. Though there are few aspects of the review that needs attention to be more useful to the audience and they are discussed below.
1. The review lacks discussion on key knowledge gaps and questions in the field. This will be of key interest to the readers. For example, from the table 1, one can clearly say that we still do not know much about the interplay between estrogen and DDR. There is a knowledge gap regarding how much mRNA upregulation of DSB repair pathway genes translate into phenotype of consequence. Also, do we know how much DSB repair gene protein expression (not mRNA) is elevated with estrogen?
2. More of a suggestion than comment is to bind different interplay between estrogen and response to DSB in one figure.
3. Another suggestion is to include a speculative discuss regarding therapeutic implication of the interplay between estrogen and response to DSB.
Author Response
We would like to thank Reviewer #2 for his/her approval of our manuscript and for the excellent comments mentioned for improving the review.
- – The review lacks discussion on key knowledge gaps and questions in the field. This will be of key interest to the readers. For example, from the table 1, one can clearly say that we still do not know much about the interplay between estrogen and DDR. There is a knowledge gap regarding how much mRNA upregulation of DSB repair pathway genes translate into phenotype of consequence. Also, do we know how much DSB repair gene protein expression (not mRNA) is elevated with estrogen?
We thank the reviewer for these comments. We added a column to Table 1 for indicating changes in the levels of proteins involved in DSB repair. We also added the knowledge gap between mRNA upregulation and phenotype consequence in the text ("Estrogen and DSB repair efficiency" part).
- – More of a suggestion than comment is to bind different interplay between estrogen and response to DSB in one figure.
We agree with the reviewer that putting together the different interplay between estrogen and DSB response in one figure is a good idea. However, since the role of estrogen in DSB repair is controversy we decided that the figure will not be clear and therefore did not add the figure.
- – Another suggestion is to include a speculative discuss regarding therapeutic implication of the interplay between estrogen and response to DSB.
Here again, we decided not to speculative discuss the therapeutic implication of the interplay between estrogen and response to DSB, since we did not want to get into the controversy regarding the effect of estrogen on DSB repair.